# Prediction of a Difficult Airway Using the ARNE Score and Flexible Laryngoscopy in Patients with Laryngeal Pathology

**DOI:** 10.3390/medicina60040619

**Published:** 2024-04-10

**Authors:** Danica Marković, Maja Šurbatović, Dušan Milisavljević, Vesna Marjanović, Biljana Stošić, Milan Stanković

**Affiliations:** 1Clinic for Anesthesiology and Intensive Therapy, University Clinical Center in Niš, 18000 Niš, Serbia; drvesnamarjanovic@gmail.com (V.M.); b.stosic@yahoo.com (B.S.); 2Clinic for Anesthesiology and Intensive Therapy, Military Medical Academy, 11000 Belgrade, Serbia; maja.surbatovic@gmail.com; 3Faculty of Medicine of the Military Medical Academy, University of Defence, 11000 Belgrade, Serbia; 4Otolaryngology Clinic, University Clinical Center in Niš, 18000 Niš, Serbia; dusanorldusan@gmail.com (D.M.); milanorlstankovic@gmail.com (M.S.); 5Department Otorhinolaryngology, Faculty of Medicine, Univeristy in Niš, 18000 Niš, Serbia; 6Department Surgery and Anesthesiology and Reanimatology, Faculty of Medicine, University in Niš, 18000 Niš, Serbia

**Keywords:** laryngoscopy, airway assessment, intubation, difficult, risk score

## Abstract

*Background and Objectives*: The ARNE score was developed for the prediction of a difficult airway for both general and ear, nose and throat (ENT) surgery with a universal cut-off value. We tested the accuracy of this score in the case of laryngeal surgery and provided an insight into its effects in combination with flexible laryngoscopy. *Materials and Methods*: This prospective pilot clinical study included 100 patients who were being scheduled for microscopic laryngeal surgery. We calculated the ARNE score for every patient, and flexible laryngoscopy was provided preoperative. Difficult intubation was assessed according to the intubation difficulty score (IDS). *Results*: A total of 33% patients had difficult intubation according to the IDS. The ARNE score showed limited accuracy for the prediction of difficult intubation in laryngology with *p* < 0.0001 and an AUC of 0.784. Flexible laryngoscopy also showed limitations when used as an independent parameter with *p* < 0.0001 and an AUC of 0.766. We defined a new cut-off value of 15.50 for laryngology, according to the AUC. After the patients were divided into two groups, according to the new cut-off value and provided cut-off value, the AUC improved to 0.707 from 0.619, respectively. Flexible laryngoscopy improved the prediction model of the ARNE score to an AUC of 0.882 and of the new cut-off value to an AUC of 0.833. *Conclusions*: It is recommended to use flexible laryngoscopy together with the ARNE score in difficult airway prediction in patients with laryngeal pathology. Also, the universally recommended cut-off value of 11 cannot be effectively used in laryngology, and a new cut-off value of 15.50 is recommended.

## 1. Introduction

A difficult airway is defined by the American Association of Anesthesiologists (ASAs) as “the clinical situation in which a conventionally trained anesthesiologist experiences difficulty with facemask ventilation of the upper airway, difficulty with intubation, or both” [1]. There are many developed clinical tools for preoperative assessment of difficult endotracheal intubation in modern anesthesiology, but no matter how detailed assessment may be, there is still a significant number of unanticipated airway difficulties in everyday clinical practice. This is why there is a need to provide further studies which would define more precise and accurate parameters and tests. The perfect preoperative test of difficult airway assessment should be cost-effective, fast, accurate and be able to timely predict serious complications [1,2,3]. 

Preoperative airway assessment is even more difficult in the ear, nose and throat (ENT) surgery, and especially laryngology, where an anesthesiologist is faced with morphological changes on airway itself. The specificity of laryngeal surgery itself implies that the anesthesiologist and the surgeon share the same working field and that the airway becomes completely inaccessible to the anesthesiologist after endotracheal intubation. The guidelines and recommendations imply that one of the main indications of a difficult airway is the presence of upper airway pathology [1]. Every patient who is planned for laryngeal surgery has upper airway pathologies of different morphology, size and location. Laryngeal pathology is very heterogenous, and every patient should be considered independently by both the anesthesiologist and surgeon. In laryngeal surgery, the airway may be compromised before, during and after surgery, and, therefore, it is recommended to have the “difficult airway” cart available in the operating room [4]. There is a significantly higher incidence of a difficult airway in ENT surgery than in other surgical fields. Wong et al. indicated that the number of patients with a difficult airway in ENT surgery is 12.6%; however, this number is even higher in laryngology [5]. The incidence of difficult endotracheal intubation in other surgical fields is as low as 0.26% [6]. The greatest challenge for both the anesthesiologist and ENT surgeon is the “cannot intubate, cannot ventilate” situation, which may result in great morbidity and mortality. These situations have greater incidence in laryngology, and this is the reason why an experienced anesthesiologist should always be present in the ENT operating room [7,8].

Over the years of practice, extensive research resulted in many prediction tools which are used in modern anesthesiology and are useful in other surgical fields. However, there are multiple clinical situations where not a single anesthesiologic prediction assessment tool, which is useful in other surgical fields, can effectively be used in ENT surgery. There is a constant need to develop specific assessment tools for ENT surgery and especially laryngology. Research in this field is scant and mostly based on case reports.

In the year of 1998, Arne et al. developed and validated a single index for the prediction of a difficult airway for both general and ENT surgery and provided a universal cut-off value [9,10]. This is the only score developed while analyzing the patients with ear, nose and throat pathology and the only score that is considered to have specificity for this pathology. Since ENT surgery has a broad spectrum of head and neck pathologies, we tested the accuracy of this score in patients who had laryngeal lesions and masses. 

With the development of medical engineering, there is an emerging trend to use modern means of visualization and diagnostics [11,12,13,14]. The expert consensus on difficult airway assessment recommended in 2023 that transnasal endoscopy can be used for preoperative assessment of patients with periglottic lesions or abnormal airway structures [11]. Since laryngeal pathology is very specific, we have tested the accuracy of a combination of flexible laryngoscopy with the ARNE score. 

## 2. Materials and Methods

We have included a total of 100 patients in this prospective pilot clinical study. All the included patients were scheduled for microscopic laryngeal surgery at the Clinic for Otorhinolaryngology, University Clinical Center in Nis, in the period from June to November 2023. The inclusion criteria were diagnosis of a vocal cord change, planned general endotracheal anesthesia, age over 18 years and the absence of a tracheostomy. The exclusion criteria were patients younger than 18 years, presence of a tracheostomy cannula, patients’ refusal to participate in our research, inability to understand and/or sign an informed consent and urgent surgical interventions. 

All the included patients were informed about our research, and they signed an informed consent. Clinical examination was performed by an anesthesiologist a day before surgery, and a specially designed questionnaire was filled. Among other general and clinical data, this questionnaire contained the data needed to calculate the ARNE score. The data needed for the ARNE score calculation are provided in Table 1. Total possible points are 48; however, according to authors’ recommendations, 11 points are used as a cut-off value. Therefore, all the patients who have a total score above 11 were considered to have potentially difficult intubation.

All the included risk factors were measured according to previous relevant recommendations and everyday clinical practice. The following pathologies were considered to be associated with difficult intubation: malformations in the facial area, acromegaly, cervical spondylosis with limited neck mobility, atlanto-occipital joint diseases, respiratory tract tumors and stiff joint syndrome. In our study, all the included patients had the initial score of 5 since one of the inclusion criteria was “change on the vocal cords”. Clinical symptoms of airway pathology were defined as one of the following: dyspnea, dysphonia, dysphagia and sleep apnea syndrome. 

Each patient was clinically examined by an ENT surgeon, and flexible laryngoscopy was performed with the patient sitting upright, leaning towards slightly. Topical local anesthesia was applied before the procedure, and flexible laryngoscope was introduced transnasally. The patient was instructed to breathe through the nose during the entire procedure. In our study, there was no need to additionally sedate the patients. After reaching the pharynx, the base of the tongue, epiglottis and surrounding structures were analyzed. The patient was then instructed to vocalize and protrude the tongue in order to visualize the lower structures of interest. Before the surgery, the ENT surgeon was asked to declare the airway as difficult or not according to the observed pathology and previous clinical experience. Even if anesthesiologists in our facility are experienced with flexible laryngoscopy, there were two reasons why this test was performed by an ENT specialist. Flexible laryngoscopy is provided as a routine ambulatory preoperative diagnostic procedure in patients with laryngeal pathology by the attending ENT surgeon. Repeating this test preoperatively, by an attending anesthesiologist, would be unnecessary. Also, ENT surgeons have valuable knowledge of the pathology itself, the type of tumor masses and their consequences on the intubation itself. It is important to mention that the surgeon was not aware of the results of previously provided tests, the measurements carried out by the anesthesiologist or the potential risk for difficult intubation determined by the anesthesiologist.

In order to exclude possible bias, endotracheal intubation was provided by an experienced anesthesiologist in the ENT field (with minimum three years in practice) who was unaware of the patient’s measured parameters and flexible laryngoscopy results. Every patient had standard monitoring, which included pulse oximetry, electrocardiogram, monitoring of non-invasive blood pressure, end-tidal carbon dioxide and airway pressure. Intramuscular premedication with midazolam and atropine was provided to each patient half an hour before entering the operating room. Propofol, fentanyl and succinylcholine were used in anesthesia induction. Because of the specificity of the pathology itself, manual ventilation of the patient was estimated before administering the muscle relaxant. Direct laryngoscopy was initially assessed using a Macintosh spatula, and the patient was intubated with a flexometallic tube of a smaller diameter. The difficulty of intubation was assessed according to the clinically accepted intubation difficulty scale (IDS). The parameters needed to calculate the IDS are provided in Table 2, and these parameters were included in the questionnaire after the intubation was completed. We considered the following as “alternative techniques”: repositioning of the patient, changing of the laryngoscope blade, change in the endotracheal tube size and use of bougie or fiberoptic bronchoscope. 

Classical endotracheal intubation with a Macintosh blade was attempted a maximum of three times; however, if the glottic view was considered to be insufficient with traditional method, an anesthesiologist would choose the most suitable alternative intubation method after the first attempt. If there existed any doubts about the success of endotracheal intubation, the surgical team was alarmed to be prepared to provide the surgical airway. If the intubation provided by an experienced anesthesiologist was not possible with fiberoptic bronchoscope, the patient was considered to be impossible to intubate and the surgical airway was established.

All results concerning continuous variables were presented as mean and standard deviation. The statistical significance was calculated using the Mann–Whitney U test for heterogenous groups and the Chi-square tests were used to determine the relationships between two categorical, independent groups, when dealing with ordinal or nominal data. We have used binary statistic regression to predict the outcome and combine different parameters. The area under the curve (AUC) was determined in order to test the performance of the tests and with the aim of defining the cut-off value of statistically significant parameters. P value below 0.05 was considered a statistically significant result. All results were statistically processed in the program SPSS10.0 (Statistical Package for the Social Sciences, Chicago, IL, USA) for Windows.

The study was approved by the Ethical Committee of Medical School, University in Niš, Niš, Serbia and by the Ethical Committee of the University Clinical Center of Niš, Niš, Serbia.

## 3. Results

The mean age of the included patients was 60.31 ± 31 years, with 37% of female and 63% of male patients. A total of 33 patients (33%) had difficult intubation according to the IDS score with 4% of patients who were not possible to intubate with any alternative technique due to changes on the vocal cords and surgical airway was established. All of the patients who were impossible to intubate had a Cormack–Lehane score of 4; two of them had extremely limited neck extension that made later surgical procedures very hard to perform. When it comes to pathological changes on the airway, one of the patients had extensive changes on the glottic and subglottic level, two of the patients had massive vegetative tumors which propagated in the subglottic area, and one of them had a flexible laryngoscopy result that did not indicate an impossible airway. This patient had a positive history of obstructive sleep apnea (OSA). 

There was a need to use alternative intubation techniques in 30 patients out of which 26 (88.66%) were difficult to intubate, according to the IDS. Details about the types of alternative techniques used are provided in Figure 1, with reference to two or more techniques used in several patients. 

After calculating the IDS, we divided the patients into two groups: the difficult intubation (DI) group and the normal intubation (NI) group. Then, we calculated the ARNE score for each patient according to the data provided in Table 1. When calculated, the mean ± SD for the ARNE score was 14.13 ± 4.58. The ARNE score showed statistical significance for predicting difficult intubation in laryngology with *p* < 0.0001. However, the calculated AUC was 0.784 (*p* < 0.0001, 95% CI 0.685–0.884), and this result indicated the limited clinical values of this score in laryngology (Figure 2). It is important to signify that as much as 85% of patients had one or more clinical symptoms of airway pathology, which adds 3 points to each of the patients. As an independent parameter, flexible laryngoscopy also showed to be accurate with *p* < 0.0001, but certain limitations of its independence were obvious in the result when the AUC was 0.766 (*p* < 0.0001, 95% CI 0.657–0.875) (Figure 2).

We used the recommended cut-off value to divide the patients into two groups, above 11 and below 11, according to which as much as 75% were defined as “difficult to intubate”. This model retained a statistical significance of *p* = 0.010, but lost statistical significance in C statistics with an AUC of 0.619 (*p* = 0.054, 95% CI 0.517–0.730). We then defined a new cut-off value of 15.50, according to the AUC ROC results, with a sensitivity of 69.7% and a specificity of 71.6%. When we divided the patients into two groups according to the new cut-off value, the new statistical model improved with *p* < 0.0001 and an AUC of 0.707 (*p* = 0.001, 95% CI 0.596–0.817) (Figure 3). Also, this new cut-off value defined 42% of the included patients as “difficult to intubate”.

Since flexible laryngoscopy is routinely used in ENT ambulatory settings and every patient with laryngeal pathology is examined preoperatively, we consider this test to be cost-effective and widely available. If not available, indirect laryngoscopy can be routinely used. We have, therefore, combined flexible laryngoscopy with the ARNE score, the ARNE score with the recommended cut-off value and the ARNE score with the new cut-off value. The results are provided in Figure 4 and Table 3. 

## 4. Discussion

The percent of difficult intubations in our study is 33%, which is higher than in other studies [6,15,16]. This can be explained by the fact that all the included patients had tumors on laryngeal level, which, in some cases, made endotracheal intubation more difficult [17]. This is confirmed by the data provided by Barclay-Steuart et al., who had an incidence of predicted difficult intubations of 24.7% in patients with pharyngolaryngeal lesions [18]. Research that included only endotracheal intubation in laryngology are scant and mostly based on case reports. The only research including patients scheduled for microlaryngoscopy was the research conducted by Tirelli et al., where the ENT specialists tended to find valuable predictors that would indicate the existence of possible difficult laryngeal exposure during the surgical procedure. However, this research did not include the ARNE score nor flexible laryngoscopy [19]. The number of impossible intubations in our study was as high as 4% of included patients, and the airway of these patients was surgically established. Research show that in our county there is a high number of patients who present with advanced disease and that the median lost time is 8.77 months beginning from the first symptoms to diagnosis [20,21]. The treatment outcomes and survival rates depend on the established stage of disease. Also, these patients are harder to intubate since they present with a high-rate stridor and a greater airway obstruction [6,22,23]. This can also be considered as the reason why as many as 85% of the included patients had clinical symptoms of airway pathology. Tasli et al. reported an incidence of 3.3% of impossible intubations in ENT patients included in their study, which correlates with our results [24]. As previously mentioned, one of the patients who was impossible to intubate had no previous indications of a difficult airway based on flexible laryngoscopy. This is one of the ongoing problems of ENT anesthesiology since certain structures collapse after the induction of anesthesia and this sometimes makes the airway impossible due to existing pathology [25]. The collapse of upper airway structures is also the main cause of obstructive sleep apnea (OSA), and it is well known that many patients who are in need of ENT surgical interventions have severe OSA and continuous positive airway pressure (CPAP) treatment as part of their medical history [26]. 

There is a dire need for an accurate score in ENT and especially laryngeal surgery. There are no official recommendations, and research in this field are rare; the generally used parameters and tests of difficult intubation prediction are not accurate in ENT surgery [27,28]. The score provided by Arne et al. is presented as specific for both general and ENT surgeries; however, specificity of laryngeal pathology implied that this score may not be completely accurate [9]. We can see in our results that the ARNE score alone cannot be used independently as a completely accurate prediction score, with an AUC of 0.784. Also, when the recommended cut-off value of 11 is provided, the AUC is statistically insignificant and falls to 0.619, with 75% of patients being predicted as difficult to intubate. One of the reasons for this can be found in the fact that all the included patients had maximum points for the risk factor named “Pathologies associated with difficult intubation” and as much as 85% of patients had maximum points for “Clinical symptoms of airway pathology”. Also, mean ± SD for the ARNE score was as high as 14.13 ± 4.58, and this alone shows the inaccuracy of the recommended cut-off value in laryngology. 

According to the AUC ROC curve, we have recommended a new cut-off value of 15.50, which improves the AUC to 0.707 and brings back the statistical significance to the model. However, even with this result, the ARNE score still cannot be considered to be independently accurate in the preoperative settings. This bring us back to the problem—what is the right mean to distinguish real difficult intubation in laryngeal pathology?

As previously mentioned, there is a growing tendency to use modern diagnostic and visual technologies in patients when there is a predicted high risk for difficult intubation. There is a special need for their use in laryngology since airway pathology can severely compromise endotracheal intubation [11,29,30]. The AUC of flexible laryngoscopy in our study was 0.766, which correlated with the results of Gemma et al. who tested the accuracy of flexible laryngoscopy in ENT patients [13]. The importance of flexible laryngoscopy in preoperative settings and the specificity of laryngeal pathology can be seen in the study conducted by Guo et al. They included the patients who were being prepared for general anesthesia in both ENT and other fields of anesthesia but excluded the patients with masses that would compromise the airway. The AUC for flexible laryngoscopy showed to be as high as 0.91 [31]. 

When added to the ARNE score without the cut-off values and with the recommended and new cut-off values, flexible laryngoscopy improves the prediction with an AUC of 0.882, 0.813 and 0.833, respectively. Flexible laryngoscopy, together with other visual technology, has improved difficult intubation prediction in other studies [13,32]. This indicates the necessity of multidisciplinary preoperative assessment of airway in laryngology, which is also confirmed by other authors in, mostly, case reports [33,34,35]. When both present, the risk factors included in the ARNE score, “Pathologies associated with difficult intubation” and “Clinical symptoms of airway pathology” results in an ARNE score of 8. This requires the use of the new recommended cut-off value of 15.50 in laryngology. Arne at al. have provided a reliable score for anesthesiology practice, which included the most frequently used preoperative tests and measurements. The fact that it includes airway pathology as a risk score makes it useful in ENT surgery. However, the specificity of the laryngeal pathology made it dependent of the surgical risk score, e.g., flexible laryngoscopy. The ARNE score, together with flexible laryngoscopy, can provide fast and accurate preoperative prediction of a difficult airway. 

There exist certain limitations of our study. The number of patients included represent a limitation, and there is a need to expand the study in the future. This limitation is primarily represented in the fact that as much as 85% of the patients had one or more symptoms of laryngeal pathology and this contributed to the limited accuracy of the ARNE score. Also, there is a great limitation in our research when it comes to the risk factor “Previous knowledge of difficult intubation”. Certain hospitals and countries have official registries of difficult intubations; however, there are no official data about previous difficult intubations in our country and many times patients are not informed about the existence of a difficult airway [36]. However, these data could only make the final ARNE score higher and result in a higher cut-off point. Also, when it comes to flexible laryngoscopy, surgeons provided subjective opinion about the existence of a difficult airway based on their extensive experience in the field. There is a developed Tasli classification of transnasal flexible laryngoscopy; however, this cannot be applied to laryngeal tumor pathology since this classification is mostly based on a normal airway and is similar to the Cormack–Lehane gradation [24]. One of the future directions of our research will be to develop a classification of flexible laryngoscopy that would be specially designed for laryngology. 

There is a necessity for further research with the aim of developing accurate tests and scores for the prediction of a difficult airway in laryngology. These models would have to include modern visual diagnostic technology and surgical assessment. 

## 5. Conclusions

The ARNE score has limited accuracy as an independent predictor in difficult airway prediction in patients with laryngeal pathology. It is recommended to use flexible laryngoscopy and a multidisciplinary approach in difficult airway prediction. Also, the universally recommended cut-off value of 11 cannot be effectively used in laryngology and a new cut-off value of 15.50 is recommended. 

## Figures and Tables

**Figure 1 medicina-60-00619-f001:**
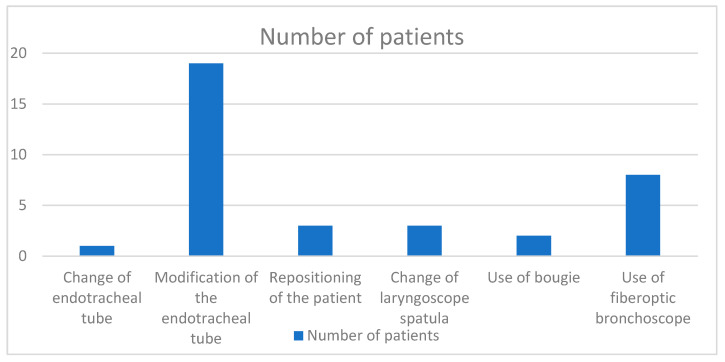
Box plot of the number of patients divided into different alternative intubation methods.

**Figure 2 medicina-60-00619-f002:**
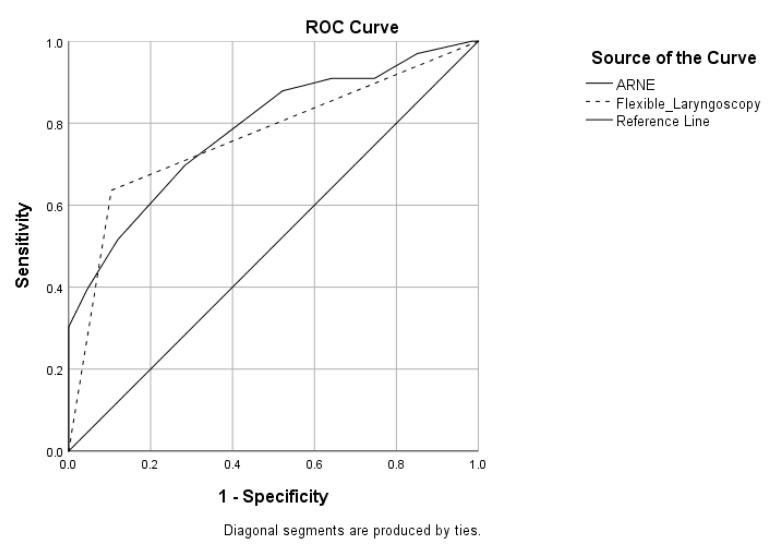
The ROC curves showing the specificity and sensitivity of the ARNE score and flexible laryngoscopy: the ARNE score with AUC = 0.784 (95% CI 0.685–0.884, *p* < 0.0001); flexible laryngoscopy with AUC = 0.766 (95% CI 0.657–0.875, *p* < 0.0001).

**Figure 3 medicina-60-00619-f003:**
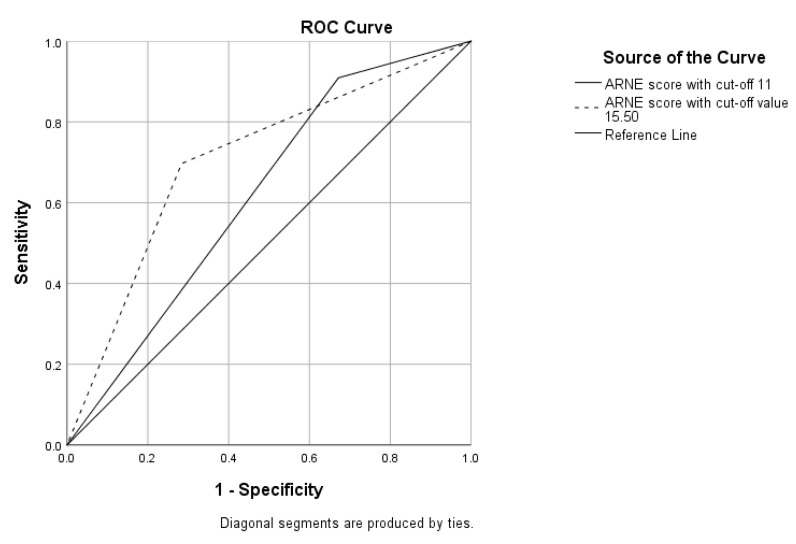
The ROC curves showing the specificity and sensitivity of the ARNE score with the official and the new cut-off values: the ARNE score with a cut-off value of 11 and AUC = 0.619 (95% CI 0.517–0.730, *p* < 0.054); the ARNE score with a cut-off value of 15.50 and AUC = 0.707 (95% CI 0.596–0.817, *p* = 0.001).

**Figure 4 medicina-60-00619-f004:**
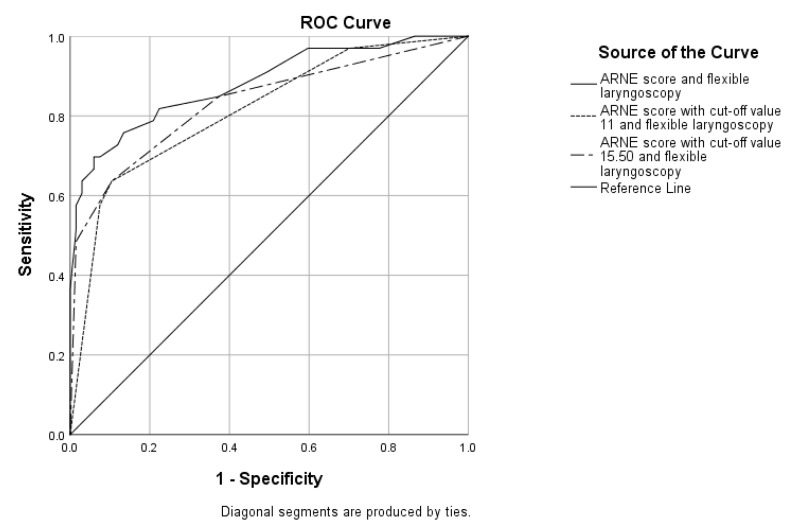
The ROC curves showing the specificity and sensitivity of combination between flexible laryngoscopy and the ARNE score with and without the cut-off values: flexible laryngoscopy and the ARNE score with AUC = 0.882 (95% CI 0.806–0.957, *p* < 0.0001); flexible laryngoscopy and the ARNE score with a cut-off value of 11 and AUC = 0.813 (95% CI 0.721–0.905, *p* < 0.0001); flexible laryngoscopy and the ARNE score with a cut-off value of 15.50 and AUC = 0.833 (95% CI 0.740–0.926, *p* < 0.0001).

**Table 1 medicina-60-00619-t001:** Calculation of the ARNE score.

Risk Factors		Points
Previous knowledge of difficult intubation	No	0
	Yes	10
Pathologies associated with difficult intubation	No	0
	Yes	5
Clinical symptoms of airway pathology	No	0
	Yes	3
Inter-incisor gap (IIG) and mandible luxation (ML)	IIG ≥ 5 cm ili ML > 0	0
	3.5 cm < IIG < 5 cm i ML = 0	3
	IIG < 3.5 cm i ML < 0	13
Thyromental distance	≥6.5 cm	0
	<6.5 cm	4
Maximal range of head and neck movement	>100°	0
	90° ± 10°	2
	<80°	5
Mallampati’s modified test	Class 1	0
	Class 2	2
	Class 3	6
	Class 4	8
TOTAL POSSIBLE		48

**Table 2 medicina-60-00619-t002:** Calculation of the IDS score.

Parameter	Score
Number of attempts > 1	Every additional attempt adds 1 point
Number of operators > 1	Every additional operator adds 1 point
Number of alternative techniques	Seach alternative technique adds 1 point
Cormack–Lehane grade	Grade 1 = 0 points
	Grade 2 = 1 points
	Grade 3 = 2 points
	Grade 4 = 3 points
Lifting force required	Normal = 0 points
	Increased = 1 points
Laryngeal pressure	No = 0 points
	Yes = 1 points
Vocal cords mobility	Abduction = 0 points
	Adduction = 1 points

**Table 3 medicina-60-00619-t003:** Detailed representation of the statistical model after applying logistic regression for the combination of the ARNE score and flexible laryngoscopy.

Parameter Added to Flexible Laryngoscopy	χ^2^	*p*	Nagelkerke R^2^	% of Correct Classification of Cases
ARNE score	57.708	<0.0001	57	86
ARNE score with old cut-off value	35.447	<0.0001	41.5	81
ARNE score with new cut-off value	40.626	<0.0001	46.5	81

## Data Availability

Data are available on demand.

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
