# Peer review of "Prediction of a Difficult Airway Using the ARNE Score and Flexible Laryngoscopy in Patients with Laryngeal Pathology"

_medicina, 2024, doi:10.3390/medicina60040619_

Round 1

Reviewer 1 Report

Comments and Suggestions for Authors
  • -There is a section required to address all limitations.

  • -How did you attempt to mitigate bias in the study?

  • -Have you considered the success rate and difficulty of intubation among individuals with varying levels of expertise and experience? If so, how?

  • -Conclusion is not acceptable
  •  

Author Response

Dear Reviewer No 1,

We thank you for your valuable comments and we have revised our manuscript according to them. We hope that this improvement would be satisfactory. We have added the following changes:

  • The section with limitations of the study has been added in the Discussion.
  • We have explained the way that we have mitigated bias in the study in the Materials and Methods section.
  • All of the anesthesiologists who were included in the study have years of practice in the ENT anesthesiology field. They are all experienced professionals and, therefore, we had no need to consider the difference between their success rate and difficulty of intubation.
  • An error has occurred while writing the article, e.g. it was not clear that ARNE score and flexible laryngoscopy both had certain limitations in independent use. The AUC below 0.8 is considered to have limited value in prediction and therefore none of them cannot be used independently in laryngology. We thank you for indicating this error to us. It is corrected in both Results and Conclusion sections.

As you will be able to see in the revised version, there have been corrections and additional text in other sections of the manuscript.

We hope that the revised version of the manuscript will be satisfactory. We thank you again for your valuable revision.

Best regards,

Authors

Reviewer 2 Report

Comments and Suggestions for Authors

This manuscript on predicting difficult airway using ARNE score and flexible laryngoscopy in patients with laryngeal pathology presents a significant contribution to the field. The prospective study design and the integration of modern diagnostic tools with established scoring systems are commendable. However, to enhance the manuscript's impact and clarity:

·        Please consider elaborating on the methodology, particularly on the selection and handling of patients who could not be intubated, to provide deeper insights into your approach and the challenges encountered

·        Consider including a more detailed discussion on the implications of your findings for clinical practice, especially regarding the new cut-off value for the ARNE score in laryngeal pathology patients

·        I recommend incorporating a comparison with existing literature, particularly focusing on the predictive value of the ARNE score and flexible laryngoscopy in different patient populations or surgical settings

·        Please include a section discussing potential limitations of your study and how these might affect the interpretation of your results

·        Consider a more detailed analysis of the statistical methods used, ensuring transparency and reproducibility of your findings

Comments on the Quality of English Language

The English language in the manuscript is generally clear and well-structured, facilitating comprehension of complex medical concepts and research findings. However, minor edits may improve occasional grammatical errors and bolster sentence structure for enhanced clarity and flow.

Author Response

Dear reviewer No2,

Thank you for you comments considering our study. It is great pleasure to receive such a review since this is the research that we have really commited to and it brought anesthesiologists and surgeons closer. We have analyzed you valuable suggestions and made certain changes in our manuscript. We hope that the changes are satisfactory.

  • We have elaborated the identification and handling of the patients who were impossible to intubate. And added the details about their pathology.
  • A more detailed discussion on the implications of our findings for clinical practice have been added, also regarding the new cut-off value.
  • There are not many research considering ARNE score, therefore, we have not been able to discuss much and compare our results. However, we have added discussion considering flexible laryngoscopy in ENT and other fields of anesthesiology.
  • We have included a section considering the limitations of the study in the Discussion section.
  • We have provided a more detailed explanation of statistical methods used and provided a Table 3 with details of statistical models.

We thank you again for your valuable comments and suggestions, which we have implemented in our manuscript.

Best regards,

Authors

Reviewer 3 Report

Comments and Suggestions for Authors

Dear Authors,

Thank you very much for your well-written manuscript, dealing with an interesting issue in anesthesiology, which is the pre-operative assessment of the difficult airway in patients with an underlying laryngeal pathology. Please pay attention to the following comments and questions, pertaining to your manuscript:

1.      A specially designed questionnaire was filled by an anesthesiologist the day before the surgery and the ARNE score was calculated. Before the surgery, each patient was clinically examined by an ENT surgeon and flexible laryngoscopy was performed. Please explain in your text: (1.) if the assessment of the second specialist was blinded and independent of the assessment of the first specialist. (2.) why different specialists (anesthesiologists vs. ENT surgeons) were used for the calculation of the ARNE score and the conduction of the flexible laryngoscopy.

2.      Please describe how flexible laryngoscopy was conducted (medication and regions of local anesthesia, trans nasal or trans oral etc.)

3.      Please provide the full term of the abbreviation ENT (ear, nose and throat) in your abstract and in the text.

Best Regards

Author Response

Dear reviewer No 3,

Thank you very much for your encouraging words considering our manuscript. We have analyzed your comments and suggestion and made the following changes in our manuscript:

  1. The ENT surgeon was not aware of the testing results provided by the anesthesiologist. This was included in the text. We have also included the reasons why ENT specialists have provided this diagnostic procedure.
  2. The procedure of flexible laryngoscopy is explained.
  3. The full term of ENT was added in both abstract and text.

We thank you again for your valuable comments and suggestions, which we have implemented in our manuscript.

Best regards,

Authors

Round 2

Reviewer 2 Report

Comments and Suggestions for Authors

This manuscript significantly improves upon the original submission with detailed methodological explanations, enhanced statistical analysis, and a more nuanced discussion of findings. The clarification provided on the limitations of the ARNE score and the rationale behind the new cut-off value are commendable.